# Synergistic Antimicrobial Activities of Combinations of Vanillin and Essential Oils of Cinnamon Bark, Cinnamon Leaves, and Cloves

**DOI:** 10.3390/foods10061406

**Published:** 2021-06-17

**Authors:** Rita Cava-Roda, Amaury Taboada-Rodríguez, Antonio López-Gómez, Ginés Benito Martínez-Hernández, Fulgencio Marín-Iniesta

**Affiliations:** 1Group of Research Food Biotechnology-BTA, Department of Food Science, Nutrition and Bromatology, University of Murcia, Campus de Espinardo, 30100 Murcia, Spain; ritacava@um.es (R.C.-R.); ataboada@um.es (A.T.-R.); 2Food Safety and Refrigeration Engineering Group, Department of Agricultural Engineering, Universidad Politécnica de Cartagena, Paseo Alfonso XIII 48, Cartagena, 30203 Murcia, Spain; antonio.lopez@upct.es (A.L.-G.); GinesBenito.Martinez@upct.es (G.B.M.-H.); 3Biotechnological Processes Technology and Engineering Lab, Instituto de Biotecnología Vegetal, Universidad Politécnica de Cartagena, Edif I + D + I, Campus Muralla del Mar, Cartagena, 30202 Murcia, Spain

**Keywords:** cinnamon, clove plant essential oils, vanillin, antimicrobial activity, combination, isobologram, synergy, *L. monocytogenes*, *E. coli* O157:H7

## Abstract

Plant bioactive compounds have antimicrobial and antioxidant activities that allow them to be used as a substitute for synthetic chemical additives in both food and food packaging. To improve its sensory and bactericidal effects, its use in the form of effective combinations has emerged as an interesting possibility in the food industry. In this study, the antimicrobial activities of essential oils (EOs) of cinnamon bark, cinnamon leaves, and clove and the pure compounds vanillin, eugenol, and cinnamaldehyde were investigated individually and in combination against *Listeria monocytogenes* and *Escherichia coli* O157:H7. The possible interactions of combinations of pure compounds and EOs were performed by the two-dimensional checkerboard assay and isobologram methods. Vanillin exhibited the lowest antimicrobial activity (MIC of 3002 ppm against *L. monocytogenes* and 2795 ppm against *E. coli* O157:H7), while clove and cinnamon bark EOs exhibited the highest antimicrobial activity (402–404 against *L. monocytogenes* and 778–721 against *E. coli* O157:H7). For *L. monocytogenes*, pure compound eugenol, the main component of cinnamon leaves and clove, showed lower antimicrobial activity than EOs, which was attributed to the influence of the minor components of the EOs. The same was observed with cinnamaldehyde, the main component of cinnamon bark EO. The combinations of vanillin/clove EO and vanillin/cinnamon bark EO showed the most synergistic antimicrobial effect. The combination of the EOs of cinnamon bark/clove and cinnamon bark/cinnamon leaves showed additive effect against *L. monocytogenes* but indifferent effect against *E. coli* O157:H7. For *L. monocytogenes,* the best inhibitory effects were achieved by cinnamon bark EO (85 ppm)/vanillin (910 ppm) and clove EO (121 ppm)/vanillin (691 ppm) combinations. For *E. coli*, the inhibitory effects of clove EO (104 ppm)/vanillin (1006 ppm) and cinnamon leaves EO (118 ppm)/vanillin (979 ppm) combinations were noteworthy. Some of the tested combinations increased the antimicrobial effect and would allow the effective doses to be reduced, thereby offering possible new applications for food and active food packaging.

## 1. Introduction

Food safety is an issue of great concern on a global scale as foodborne illnesses continue to be one of the main causes of morbidity and mortality [1]. Natural antimicrobials are considered an alternative to chemical additives, which are increasingly associated with negative health consequences, such as carcinogenicity, teratogenicity, and toxicity, as well as environmental problems related to their long periods of degradation [2,3]. 

Aromatic plants, spices, and more specifically their essential oils (EOs) have a high content of bioactive compounds and have been widely used for their bactericidal, virucidal, and fungicidal applications in various fields, including food technology (food and food packaging), medical, pharmaceutical, public health, and environmental [4]. The complex structure of EOs and the variety of chemical nature of their components are responsible for a broad range of biological interactions, many of which are of increasing interest regarding the subject of food preservation [5,6]. However, the application of EOs in food can be limited by the following factors: (a) the need for high concentrations to achieve bacteriostatic (the inhibition of bacterial growth without killing cells) or bactericidal (the destruction of bacterial cells) effects; (b) adverse effects after EO treatment (e.g., changes to the physicochemical and sensory characteristics of the subject of application); and (c) increase in costs due to higher concentrations of essential oils [7]. New ways of applying EOs are currently being studied to overcome these problems. The application of EOs in food packaging materials and coating films has been studied, along with their use directly in the food matrix as emulsions or nanoemulsions among other solutions [8].

The majority of studies regarding antibacterial complexes have aimed to achieve synergistic effects through combined treatments. Generally, drug combinations have proven to be an essential feature of antimicrobial and antioxidant treatment due to a number of important considerations: (1) they increase activity through the use of compounds with synergistic or additive activity; (2) they thwart drug resistance; (3) they decrease required doses, thereby reducing both cost and adverse/toxic side effects; and (4) they increase the spectrum of activity [9]. However, combinations of EOs can lead to synergistic, additive, or antagonistic effects. Such knowledge could contribute to designing new, safer, and more effective novel natural antimicrobial and antioxidant blends for preserving foods [10]. 

*Listeria monocytogenes* and *Escherichia coli* O157:H7 are two foodborne pathogens responsible for causing a great number of diseases with significant effects on human health and the economy [11,12,13]. Vanillin and EOs of cinnamon and clove have shown antimicrobial activity in vitro and in food products against *L. monocytogenes* [14,15,16] and *E. coli* O157:H7 [17,18,19,20]. Vanillin has been applied in several foods as an antimicrobial against *E. coli* and *Listeria innocua* [7]. Vanillin has also applied in combination with thermal treatment in milk against *L. monocytogenes* [21] and in infant formulas against *Cronobacter sakazakii* and *Salmonella enterica* subsp. *enterica* serovar. Typhimurium [7]. Vanillin has also been studied in apple juice to control *L. monocytogenes* and *E. coli* O157:H7 [22,23] and *L. innocua*. EOs of cinnamon bark and leaves in combination have been used as antimicrobials in several broths against *E. coli* O157:H7, *Salmonella enterica* subsp. *enterica* serovar. Typhimurium, and *L. monocytogenes* [7]. Clove EOs have been applied as antimicrobials in several foods, such as meat and cheese [7] or salmon burger [24]. The antimicrobial activity of the combination of clove and cinnamon essential oils against *L. monocytogenes* has been studied in celery against *L. monocytogenes* and *Salmonella enterica* subsp. *enterica* serovar. Typhimurium [14] and in pasteurized milk against *L. monocytogenes* [16]. EOs of cinnamon leaves have been applied as antimicrobials in orange and pomegranate juices [25]. Moreover, EOs of cinnamon bark and leaves and CA have been used as antimicrobials in films for active food packaging [7,26]. VA and cinnamon EOs have been applied to edible films for food packaging [27].

There have been precedent studies on the application of a combination of EOs of cinnamon and vanillin in milk [17] and combination of EOs of cinnamon bark/cinnamon leaves in celery [14]. However, no comprehensive study has been conducted to evaluate the antimicrobial properties of binary combinations of vanillin and cinnamon EOs. This work was designed to investigate whether there are synergistic antibacterial interactions of binary combinations between the EOs of cinnamon bark (CBEO), cinnamon leaves (CLEO), and clove (CLOEO) and pure plant compounds vanillin (VA), eugenol (EU), and cinnamaldehyde (CA) against *L. monocytogenes* and *E. coli* O157:H7.

## 2. Materials and Methods

### 2.1. Microorganisms and Culture Conditions

The *L. monocytogenes* Scott A strain, a human epidemic isolate, was supplied by the FDA (USA), and the *E. coli* O157:H7 5947 strain (nontoxigenic) was supplied by the Spanish Type Culture Collection, University of Valencia, Spain. All strains were kept at −80 °C in Microbank^TM^ vials (PRO-LAB Diagnostics, Neston, Wirrall, UK). Every two months, one of the vials was opened, and the stock culture was grown in Trypticase soy broth (TSB) (Cultimed, Barcelona, Spain) for 24 h at 35 °C and streaked onto PALCAM agar plates (Merck, Darmstadt, Germany), which were incubated for 48 h at 35 °C. One colony obtained in PALCAM agar was transferred to TSB and incubated for 28 h at 35 °C before being stored at −20 °C in a solution of 40% TSB and 60% glycerol until use. The fresh cultures for the experiments were made by incubating one loopful of pure culture in TSB for 24 h at 35 °C. The inocula were standardized in TSB until an optical density (OD) of 0.1 at 595 nm was reached and then diluted in TSB to obtain a bacterial concentration between 5 and 5.7 log CFU/mL. The inocula bacterial populations were counted by spreading suitable diluted aliquots onto Trypticase soy agar (TSA) plates, followed by incubation at 37 °C for 24 h.

### 2.2. Stock Antimicrobial Solutions

Vanillin (VA), eugenol (EU), and cinnamaldehyde (CA) were assayed as plant antimicrobial pure compound. Clove EO (CLOEO), cinnamon bark EO (CBEO), and cinnamon leaf EO (CLEO) were assayed as plant EOs with antimicrobial properties. VA, EU, and CA were obtained from Aldrich (Steinheim, Germany). CLOEO, CBEO, and CLEO were obtained from Destilerías Muñoz Gálvez (Murcia, Spain). The main compounds in CBEO were CA (70.6%) and EU (4.2%), those in CLEO were CA (1.2%) and EU (75.9%), and those in CLOEO were EU (76.0%) and CA (<1%).

Stock solutions were prepared by adding suitable amount of antimicrobial VA, EU, CA, CLOEO, CBO, or CLEO to TSB with Tween 80 0.5% (volume (v) (Aldrich Steinheim, Germany). In the case of VA, TSB was previously warmed at 50 °C according to Delaquis et al. [28]. The pH was adjusted to 7 with NaOH 0.1 N, prior to sterilization by filtration through 0.22 μm membranes.

### 2.3. Determination of the Minimum Inhibitory Concentration (MIC) and the Minimal Bactericidal Concentration (MCB) 

The effect of different concentrations of VA, CA, EU, CLOEO, CBEO, and CLEO on the growth of *L. monocytogenes* and *E. coli* O157:H7 was evaluated by the broth microdilution method [29]. The optical density (OD) values were obtained in TSB adjusted to pH 7 and amended with 0.15% agar as suggested by Mann and Markham (1998) [30] using a microplate OD reader (Multiskan Ascent, Thermo Fisher Scientific, Madrid, Spain) and 96-well microplates (12 columns and 8 rows A to H). The bacterial growth was associated with changes in the OD.

Five stock solutions with different concentrations were prepared in TSB + 0.30% agar for each antimicrobial as indicated above. Five wells of column one of the microplate received 300 μL of TSB + 0.30% agar as sterile control, and the other five wells of column two received 150 μL TSB + 0.30% agar and 150 μL of inoculum as positive control. To the last wells (row H) of each remaining column 3–12, 300 μL of each stock solution was added as duplicate (two columns for each stock solution). All other wells of columns 3–12 (rows E to A) received 150 μL TSB + 0.30% agar, while 150 μL of each stock solution was added to the next well in the appropriate column and mixed, thus obtaining a two-fold dilution series. Each well, except for the controls, received 150 μL of the standardized inoculum to provide an initial population close to 5 log CFU/mL. The microplate was then incubated at 35 °C for 24 h, and the OD of each well was registered every hour. The initial OD value was subtracted from the new OD values of the same well in order to only correlate the OD variation with the microorganism growth. The MIC was defined as the lowest concentration of the antimicrobial agent that produced an OD variation lower than 0.1. Each microplate was made in triplicate.

To determine the MBC of the antimicrobials, dilutions above their previously calculated MIC were prepared in test tubes containing TSB and tested against *E. coli* O157:H7 and *L. monocytogenes* populations about 5 log CFU/mL. The test tubes were incubated at 35 °C for 24 h, and 100 μL of each tube was plated on TSA plates after incubation. The MBC was defined as the lowest concentration at which no growth was observed when 0.1 mL of the sample was plated (<10 CFU/mL) [30].

### 2.4. Evaluation of Interaction between the Antimicrobials Tested

Interactions of EOs were determined using a checkerboard microdilution test as described by Nikkhah et al. [31]. The checkerboard assays were performed by mixing two compounds to determine what antimicrobial interactions could be observed if different concentrations (different proportions at MICs) were combined. Seven treatments comprising CLOEO/CLEO, CLEO/CBEO, CLOEO/CBEO, CLOEO/VA, CLEO/VA, CBEO/VA, and EU/CA were set up for the dual combinations and tested against *L. monocytogenes* and *E. coli* O157:H7. The evaluated concentrations were in the range of 10 dilutions below the MIC to twice the MIC (see Table 1). Growth control wells (medium with inoculum but without EOs) were included in each microplate. Each test was done in duplicate.

The fractional inhibitory concentration (FIC) was calculated for the first clear well in each row of the microtiter plate using the following formula:$$FIC_{A}=\frac{MIC_{\left(A\;in\;the\;presence\;of\;B\right)}}{MIC_{\left(A\;alone\right)}}$$
$$FIC_{B}=\frac{MIC_{\left(B\;in\;the\;presence\;of\;A\right)}}{MIC_{\left(B\;alone\right)}}$$

Subsequently, the *FIC* index (*FICi*) was calculated with the *FIC* for individual antimicrobials as follows:$$FICi={\mathrm{FIC}}_{A}+{\mathrm{FIC}}_{B}$$

The *FICi* of each combination of antimicrobials was obtained by calculating the *FIC* of the most effective concentrations of antimicrobials [32]. The obtained results were interpreted as follows: synergistic effect (*FICi* ≤ 0.5), additive effect (0.5 < *FICi* ≤ 1), no interactive effect (1 < *FICi* ≤ 4), and antagonistic effect (*FICi* > 4) [33,34].

The isobologram method is a graphical representation of interactions and is formed by selecting a desired FIC and plotting the individual antimicrobial doses required to generate that FIC on their respective x- and y-axes. A straight line is then drawn to connect the points. Data points below or equal to the 0.5 line in the isobologram are regarded as synergistic (area A), points between 0.5 and including 1.0 are regarded as additive (area B), points above 1.0 and including 4.0 are regarded as indifferent (area C), and points above 4.0 are regarded as antagonistic (area D) (Figure 1) [35,36].

## 3. Results

### 3.1. MIC and MBC of EOs and Vanillin

The antimicrobial activity of the CBEO, CLEO, and CLOEO and the pure compounds EU and CA against *L. monocytogenes* and *E. coli* O157:H7, which were studied using the broth microdilution method, is shown in Table 1.

The pure compound that showed the highest antimicrobial activity was CA with MICs of 449 ppm against *L. monocytogenes* and 455 ppm against *E. coli* O157:H7. Hawkins [36] found a lower MIC against *L. monocytogenes* (250 ppm), which could be attributed to the different strains of *L. monocytogenes* used. The MIC of CA against *E. coli* found by other authors was between 400 [37,38] and 500 [39] ppm.

EU presented MICs of 562 ppm for *L. monocytogenes* and 766 ppm for *E. coli* O157:H7, values that correspond to those found by other authors [40,41,42]. Although the MIC of CA was similar to that of EU for both microorganisms, the bactericidal activity of CA, which had MBC values of 1400 ppm for *L. monocytogenes* and *E. coli* O157:H7, was much lower than for EU, which had MBCs of 1000 ppm for *L. monocytogenes* and 1200 ppm for *E. coli*. The same effect was found by Gill and Holley [41], who attributed it to the mechanisms of CA and EU involved in the inhibition of energy generation and membrane permeability at bactericidal concentrations.

One of the EOs with the highest antimicrobial activity against *L. monocytogenes* was CBEO with a MIC of 404 ppm. The MIC values reported by other researchers for this EO and against this microorganism were 250 [39], 500 [19,42,43,44], 640 [45], and 1250 [46] ppm. The MIC of the CBEO found in this work for *L. monocytogenes* was somewhat lower than the MIC of CA (449 ppm), its majority component. The same effect was also found by Chang et al. [39], which was attributed to a synergistic effect between the minority components present in the CBEO. In the same way, this EO exhibited greater bactericidal activity than CA, with MBC values of 1200 versus 1400 ppm, respectively, which could also be attributed to interaction with the minor components of CBEO.

The MIC value found for the CBEO against *E. coli* O157:H7 (720 ppm) was between the range found by other authors of 250 [39,47], 500 [19], and 1000 [48] ppm. The MBC value for CBEO was also in the range (1000–1500 ppm) found in the literature [44,49]. However, the effect of minority compounds of CBEO was probably antagonistic against *E. coli* O157:H7 as the MIC (720 ppm) and the MBC (1500 ppm) were greater than that of CA, its main component (454 and 1400 ppm, respectively).

The MICs of CLEO (508 ppm) and CLOEO (402 ppm) against *L. monocytogenes* were both lower than the MICs of EU (562 ppm), the main compound of both EOs. CLEO and CLOEO exhibited MICs in the range of concentrations found by others authors for *L. monocytogenes* (400–500 ppm) [19,44,48,50,51] and *E. coli* (500–1000 ppm) [19,52]. The CLOEO presented greater antimicrobial activity than CLEO despite both having the same percentage of EU. This effect was attributed to a potential synergy between the minor components of EOs, which are present in a higher proportion in CLOEO (β-caryophyllene 17.76% or α-caryophyllene 1.94%) than in CLEO (β-caryophyllene 3.48%, benzyl benzoate 2.89%, and eugenyl acetate 1.89%). In the case of *E. coli* O157:H7, the minority components of both EOs apparently had no effect on the antimicrobial activity of CLOEO and CLEO as their MICs (843 and 778 ppm, respectively) were very similar to the MIC of the pure compound EU (766 ppm). The values were also in the range found by other authors (500–1000 ppm) [43,51]. Possible synergistic and antagonistic effects between the minor compounds of CLOEO and CLEO were also found when the MBC of both oils was studied. The MBC of CLOEO for *L. monocytogenes* (800 ppm) was lower than the MBC of EU (1000 ppm), which could be explained by a synergistic effect with the minor components of CLOEO. However, the MBC of CBEO against *E. coli* O157:H7 (1500 ppm) was higher than the MBC of EU (1200 ppm), which could be explained by an antagonistic effect with minor components of this EO.

Among the compounds tested, VA had the lowest antimicrobial activity against *E. coli* O157:H7 and *L. monocytogenes*. However, *E. coli* O157:H7 was more susceptible than *L. monocytogenes*, with MICs of 3000 and 2800 ppm, respectively. Fitzgerald et al. [53], Moon et al. [48], and Corte et al. [49] also found that *E. coli* O157:H7 was more susceptible to VA than some Listeria strains. The higher sensitivity of *E. coli* O157:H7 could be explained by the presence of certain proteins in the external membrane, which create channels for the penetration of compounds with low molecular weight [54]. 

The MBCs for VA were 8000 ppm for *L. monocytogenes* and 6000 ppm for *E. coli* O157:H7, and these values were much higher than their respective MICs. Fitzgerald et al. [55] found bactericidal activity of VA against *E. coli* O157:H7 and *L. innocua* of 6600 and 9240 ppm, respectively. According to them, the high difference between MIC and MBC can be attributed to the fact that VA has a bacteriostatic and nonbactericidal action, which is in contrast to the more potent phenolic antimicrobials such as EU, carvacrol, and thymol, which are bactericidal.

### 3.2. FICi of Binary Combinations Using Two-Dimensional Checkerboard Method 

The FICi of selected dual combinations of the antimicrobial compounds tested in this study against *L. monocytogenes* and *E. coli* O157:H7 are presented in Table 2 and Table 3, respectively.

The mixtures CBEO/CLEO and CBEO/CLOEO showed FICi minimal values against *L. monocytogenes* of 0.66 and 0.70, respectively (Table 2), which qualifies these combinations as additive. The isobolograms of both mixtures (Figure 2b,c) were located in the B area. Purkait et al. [56] found even more synergistic effects in the combinations of CBEO and CLOEO with FICi = 0.42 for these EOs, which could be attributed to the different composition of the EOs used.

The minimal FICi of the EU/CA mixture against *L. monocytogenes* was 0.75 (Table 2). The pure compounds EU and CA are the major components of the mixtures of CBEO with CLOEO and CLEO and also proved to have an additive effect with an isobologram located in the B area (Figure 2g).

The mixtures of CBEO/CLOEO and CBEO/CLEO presented FICi minimal values of 1.11 and 1.00, respectively, against *E. coli* O157:H7 (Table 3), with the position in the C area of the isobologram, qualifying these combinations as indifferent (Figure 3b,c). These FICi values indicate lower antimicrobial activity of these combinations against *E. coli* O157:H7 than *L. monocytogenes*. They were also higher than the FICi values found for the mixtures of EU and CA against *E. coli* O157:H7 (Figure 3g), whose interaction was additive and was located in the B area of the graph. Pei et al. [37] found similar results when testing mixtures of EU and CA against *E. coli*.

According to these results, the interactions between the minority compounds of the EOs had a different effect on the antimicrobial activity of the mixtures CBEO/CLOEO and CBEO/CLEO against *E. coli* O157:H7 (antagonism) and *L. monocytogenes* (synergy). Similar results were found by Lu et al. [57] with mixtures of CBEO and CLOEO, which presented an additive effect for Gram-positive bacteria such as *Bacillus cereus*, *B. subtilis*, and *Staphylococcus aureus* and an indifferent effect against Gram-negative bacteria such as *E. coli* and *S. enterica* subsp. *enterica* serovar. Typhimurium. The combination of CLEO/CLOEO for *L. monocytogenes* produced an additive effect due to its minimal FICi of 0.85 (Table 2), and its isobologram (Figure 2a) was positioned in area B of the graph. The same combination presented a minimal FICi of 0.92 against *E. coli* O157:H7 (Table 3), which was higher than that of *L. monocytogenes* but would still qualify the mix as additive although less inhibitory. The isobologram was also located in the B area (Figure 3a). The composition of both EOs is similar, with a high content in EU and very low content of CA, which might explain why the CLOEO/CLEO mixtures had less antimicrobial activity than the CLOEO/CBEO and CLEO/CBEO mixtures. The lower activity of the CLOEO/CLEO mixtures against *E. coli* O157:H7 could be explained by the fact that this microorganism is more resistant to EU than *L. monocytogenes* as well as the fact that minor components of both EOs can interact antagonistically to decrease the antimicrobial activity of EU.

The combinations of CLOEO/VA, CLEO/VA, and CBEO/VA presented minimal FICi values for *L. monocytogenes* of 0.53, 0.67, and 0.51, respectively, which would qualify them as additive, although the mixtures of VA with CLOEO or CBEO were very close to synergy (Table 2). The isobolograms were located in the B area (Figure 2d–f). 

The combinations of CLOEO/VA, CLEO/VA, and CBEO/VA against *E. coli* O157:H7 were all synergistic with minimal FICi values between 0.48 and 0.49 (Table 2), and the isobolograms were located in the C area (Figure 3d–f). These results agree with those obtained in the individual MIC tests as *E. coli* O157:H7 was also somewhat more sensitive to VA than *L. monocytogenes*.

## 4. Discussion

The antimicrobial activity displayed by CLEO, CLOEO, and CBEO can be assigned to their high content in EU or CA, as reported by some authors [58,59,60] but influenced by the minor components and the microorganism target. 

The high antimicrobial activity of CA has been demonstrated by some authors [36,61,62] and is fundamentally attributed to the aldehyde group of the molecule that is reactive and has the ability to cross-link covalently with DNA and proteins through amino groups, thereby interfering with normal cell function. It is believed that there are three reactions that potentially occur. At small concentrations, CA inhibits various enzymes involved in cytokinesis or less important cell function. At larger but sublethal concentrations, CA acts as an ATPase inhibitor. At larger but lethal concentrations, CA disrupts the cell membrane [9]. The antimicrobial activity of the EU is due to its phenol group, which coagulates protein and disrupts cell walls and cell membranes [42,60].

The antimicrobial activity of VA is due to the aldehyde group and the side group of the benzene ring. Due to its hydrophobic nature, the antimicrobial mechanisms of VA are mainly based exclusively on its ability to destroy the cytoplasmic membrane of microbial cells through interaction with lipids and proteins or with both structures, with the subsequent loss of gradient ionic and inhibition of respiratory activity [51].

Numerous studies have shown that the interaction between the components of EOs is complex and has a decisive influence on the final activity of the EO. Although the final antimicrobial activity is very close to the activity of the main component in some cases, it has been shown that a high number of compounds exhibit different antimicrobial properties when tested separately [59]. The interaction of multiple antimicrobial agents (not only within an EO but in various combinations) can result in various combined effects according to the composition and concentration of the components: (1) synergistic effect, where the antimicrobial activity of the blend of antimicrobial is greater than the sum of the effects of the individual components; (2) additive effect, where the antimicrobial activity is equal to the sum of the effects of the individual components; and (3) antagonistic effect, where the antimicrobial activity is less than the sum of the effects of the individual components [7]. 

Most studies attribute additive and synergistic effects to the combination of phenolic compounds such as EU and CA, which enhance their antimicrobial activity in mixtures mainly because they have different sites of action in the microbial cell [35,56]. The mechanisms of interaction that produce synergism include sequential inhibition of a common biochemical pathway, inhibition of different protective enzymes, combination with active agents in the cell wall that allow the entry of other antimicrobials, or interaction with the plasmatic membrane [60]. Furthermore, the main mechanism of action can be enhanced by others that are less effective and vice versa. 

In this study, the strong antimicrobial effect of CBEO/CLOEO, CBEO/CLEO, and CLOEO/CLEO mixtures is mainly attributed to the interaction between their main constituents EU (which acts at the membrane level) and CA (which inhibits the mechanisms aimed at obtaining energy), with both compounds interacting with different enzymes or proteins [61]. The greater antimicrobial effect of the mixtures in relation to the individual compounds is also explained by the presence of minor antimicrobial compounds, which together can enhance or complement their effect. 

For many plant species, there is a great difference in the chemical composition between various organs of the same plant. For this reason, the authors of [35] suggest the investigation of possible synergies combining plant extracts obtained from different parts of the same plant. Antimicrobial combinations of cinnamon bark and leaves (CBEO/CLEO) have been applied in several broths [7] and celery [14]. In our study, using the checkerboard and isobologram methods, we successfully assayed the combination EOs of CBEO/CLEO, which have very different chemical compositions. Thus, we were able to improve the antimicrobial effect because the variety of bioactive compounds provided more possibilities for synergistic interactions. In general, when plants have a similar composition, their combination has an additive effect rather than a synergistic one because they have similar sites of action, as in the case of mixtures of CLOEO and CLEO, whose main component is EU. Likewise, the combination of aromatic compounds with different chemical structures can improve their antimicrobial effect by attacking different target sites in the bacterial cell at the same time [62,63,64]. For this reason, the mixtures of VA with the EOs or the combination between CLOEO with CBEO exhibited the highest antimicrobial activity in this research. 

The mixtures CBEO/CLOEO and CBEO/CLEO showed some antagonistic effect against *E. coli* O157:H7. The mechanisms that produce antagonism are less well known, although they include combinations of bacteriostatic and bactericidal agents that act on the same target site of the microorganisms and neutralize the interactive effect between the active components of the fractions. This may block active groups of fractional components and enhance antagonism [65]. The different morphology between *L. monocytogenes* and *E. coli* O157:H7 also affect the mechanism of action of the EOs because the absence of the additional outer membrane and porins on the external peptidoglycan layer makes *L. monocytogenes* less resistant to antimicrobial agents than *E. coli* O7:H157. Generally, Gram-negative bacteria are less sensitive to antimicrobials because of the lipopolysaccharide outer membrane of this group, which restricts diffusion of hydrophobic compounds [66].

Combinations of VA and EOs (CLOEO/VA, CLEO/VA, and CBEO/VA) with synergistic antimicrobial effect (FICi values less than 0.5) or FICi values close to 0.5 have potential food applications as they allow an increase in antimicrobial activity of the VA and a decrease in its sensory impact. These combinations may be applied in various types of food and food packaging. For example, in milk, they can be used to improve the results obtained in inhibiting growth of *L. monocytogenes* and *E. coli* O157:H7 [17] or combined with heat treatment to control *L. monocytogenes* [21]. VA has been applied in apple juices to control *L. monocytogenes, E. coli* O157:H7, and *L. innocua* [22,23], so the application of CLOEO/VA, CLEO/VA, and CBEO/VA combinations of greater antimicrobial effect can be used to increase its effectiveness without negative sensory effect. VA can be used in food packaging [26], so the combinations of VA with the EOs studied herein (CLOEO/VA, CLEO/VA, and CBEO/VA) can be used in active packaging that is in direct contact with food. VA has also been used in edible packaging for apple cuts [27], so CLOEO/VA, CLEO/VA, and CBEO/VA combinations can have potential applications in this field. 

Regarding the studied combinations of CLEO/CLEO, CLEO/CLEO, and CLEO/CLEO with FICi values greater than 0.5 and less than 1 or close to 1, they can be used in various foods and food packaging. CBEO and CLEO have been used as antimicrobials in films for food packaging [7], so some of the combinations studied, such as CLEO/CBEO and CLEO/CLOEO, can also be applied in active packaging due to their greater antimicrobial effect than EOs alone. Applications of the studied combinations are also possible in pomegranate and orange juices [25] as well as in fish products [24], which are discussed below due to their importance to the sensory aspect.

The high sensory impact of plant EOs in food products often limits their application to typically spicy or flavored foods. Therefore, synergistic interactions should be used to reduce the organoleptic impact and thus facilitate application to a wider range of foods [9]. VA is used as a flavoring in numerous foods [26]. In previous studies, VA has been applied to milk at sensory acceptable concentrations of 900, 1400, and 1800 ppm with a partially inhibitory antimicrobial effect but with good results in combination with thermal treatment against *L. monocytogenes* [21]. The combinations of VA with CLEO and CBEO studied in the current work will make it possible to increase the antimicrobial effect in combination with heat treatment, thus maintaining its sensory acceptance. In addition, certain combinations of VA, CLEO, CBEO, and CLOEO have been tested in milk [17], and the binary combinations of VA (1250 ppm)/CBEO (500 ppm) and VA (725 ppm)/CBEO (500 ppm) had a particularly good antimicrobial effect and good sensory acceptance. In the current study, the best inhibitory effects were achieved by CBEO (85 ppm)/VA (910 ppm) with FICi of 51 for *L. monocytogenes* and by CBEO (144 ppm)/VA (784 ppm) with FICi of 49 for *E. coli* O157:H7. These combinations would allow a reduction of the concentrations of EOs. The sensory impact would be much lower, but the inhibitory capacity against these microorganisms would be maintained. It should also be noted that cinnamon and vanilla spice mixes have a flavor that is appreciated in many dessert recipes. Sanchez et al. [25] studied the sensory effect of CLEO in pomegranate and orange juices. They determined that a dose of 40 ppm of CLEO was rejected by all the panelists, with these observations related to significantly negative changes in aroma and flavor of both juices. The maximum concentration accepted by panelists (20 ppm) was a subinhibitory dose, but it was used successfully in combination with mild heat and ultrasound for the control of *Saccharomyces cerevisiae* in these two juices. The results of the current study open the possibility of using CLOEO/CLEO, CLEO/CBEO, and CLOEO/CBEO combinations below the sensory threshold in this type of juices, accompanied by mild heat treatment and ultrasound to achieve an improvement of the antimicrobial effect. Jonusaite et al. [24] applied CLEO for the preservation of salmon burgers at a sensory accepted limit dose of 100 ppm, so it would be possible to use combinations of CLEO/CBEO and CLEO/CLOEO to increase the antimicrobial and antioxidant effect while reducing sensory impact.

Cho et al. [7] recommended the application of combinations of essential oils with a synergistic antimicrobial effect in order to reduce the dose of essential oils applied to food. This would not only lower the sensory impact but also lower the cost of essential oils applied to achieve the same preservative effect in food.

## 5. Conclusions

The antimicrobial activity of CBEO, CLEO, CLOEO, and VA and the pure compounds EU and CA were tested individually and in combination against *L. monocytogenes* and *E coli* O157:H7. Based on the MIC values, the CBEO and CLOEO exhibited the highest antimicrobial activity against both microorganisms, but *E. coli* O157:H7 was more resistant than *L. monocytogenes*. VA was the antimicrobial with the lowest activity, with MIC values very similar for both microorganisms. For *L. monocytogenes*, pure compound EU, the main component of CLEO and CLOEO, showed lower antimicrobial activity than EOs, which was attributed to the influence of the minor components of the EOs. The same was observed with CA, the main component of CBEO. Checkerboard assays for the combination of EOs showed additive effect against *L. monocytogenes*. However, the combination of CBEO/CLOEO and CBEO/CLEO exhibited an indifferent effect against *E. coli* O157:H7, which was attributed to the influence of the minor components of the EOs. The FICi indicated that the combination of VA/CLOEO and VA/CBEO showed the most synergistic antimicrobial effect on *L. monocytogenes* and *E. coli* O157:H7. This study demonstrates that a combination of compounds from different areas of the plant, and especially from different plants, increases the antimicrobial effect and allows the effective antimicrobial dose to be reduced. For this reason, this field of synergistic studies on the combinations of plant antimicrobials must be encouraged, including a broader scope involving other microorganisms with possible new applications in food and active food packaging.

## Figures and Tables

**Figure 1 foods-10-01406-f001:**
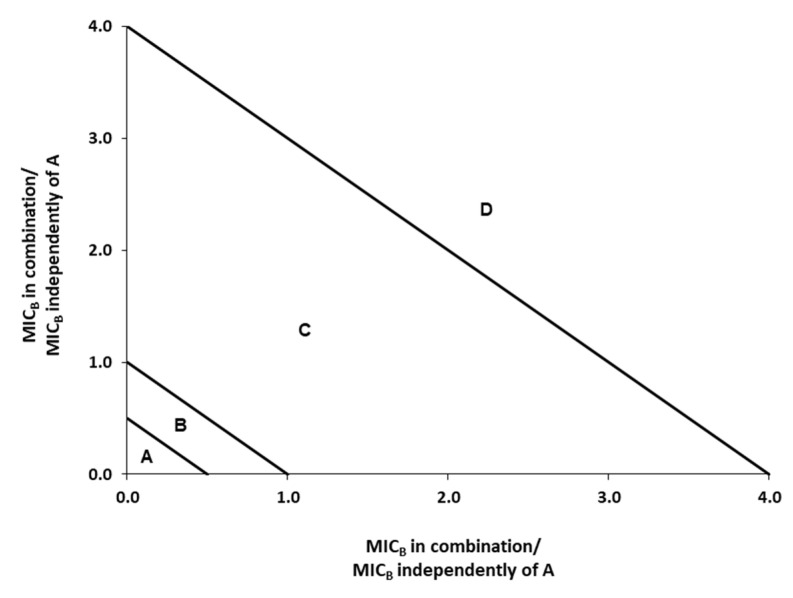
Interpretation of an isobologram. Depending on the area where the CIF values of the mixture are located, the effect is A: synergistic, B: additive, C: indifferent, or D: antagonistic [28].

**Figure 2 foods-10-01406-f002:**
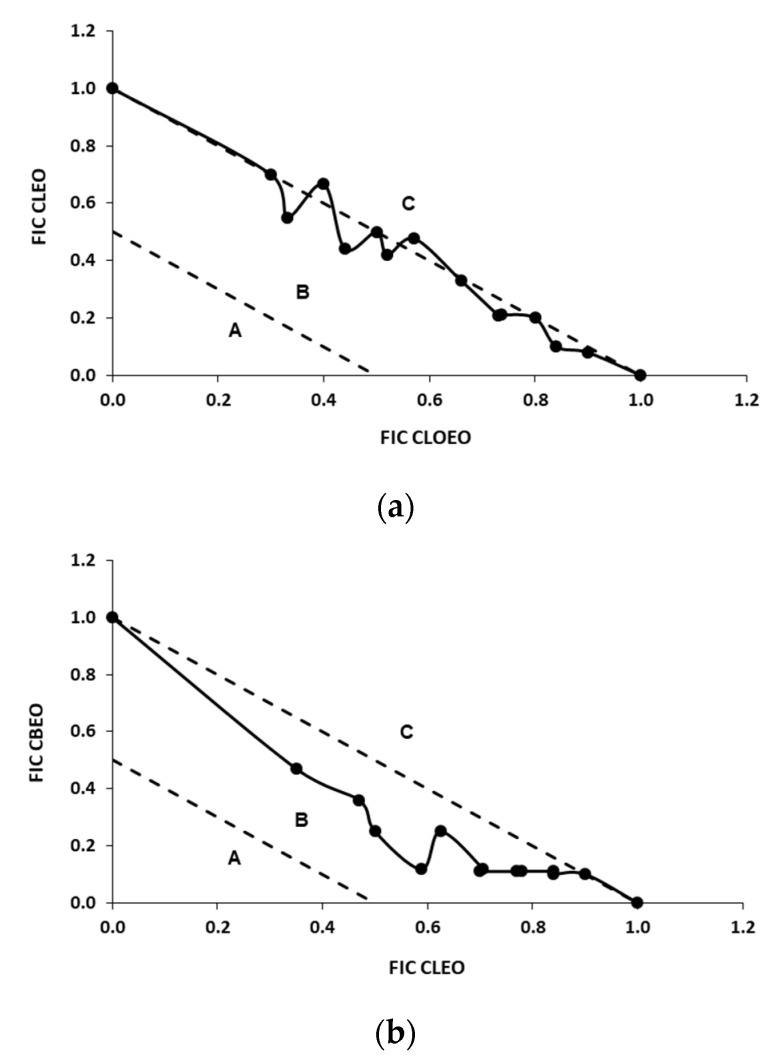
Isobolograms of selected combinations of EOs (CLEO, CBEO, and CLOEO) and pure compounds (VA, CA, and EU) against *L. monocytogenes*. (**a**) CLEO/CLOEO; (**b**) CBEO/CLEO; (**c**) CBEO/CLOEO; (**d**) CLOEO/VA; (**e**) CLEO/VA; (**f**) CBEO/VA; (**g**) EU/CA. See Figure 1 for the interpretation of the isobolograms.

**Figure 3 foods-10-01406-f003:**
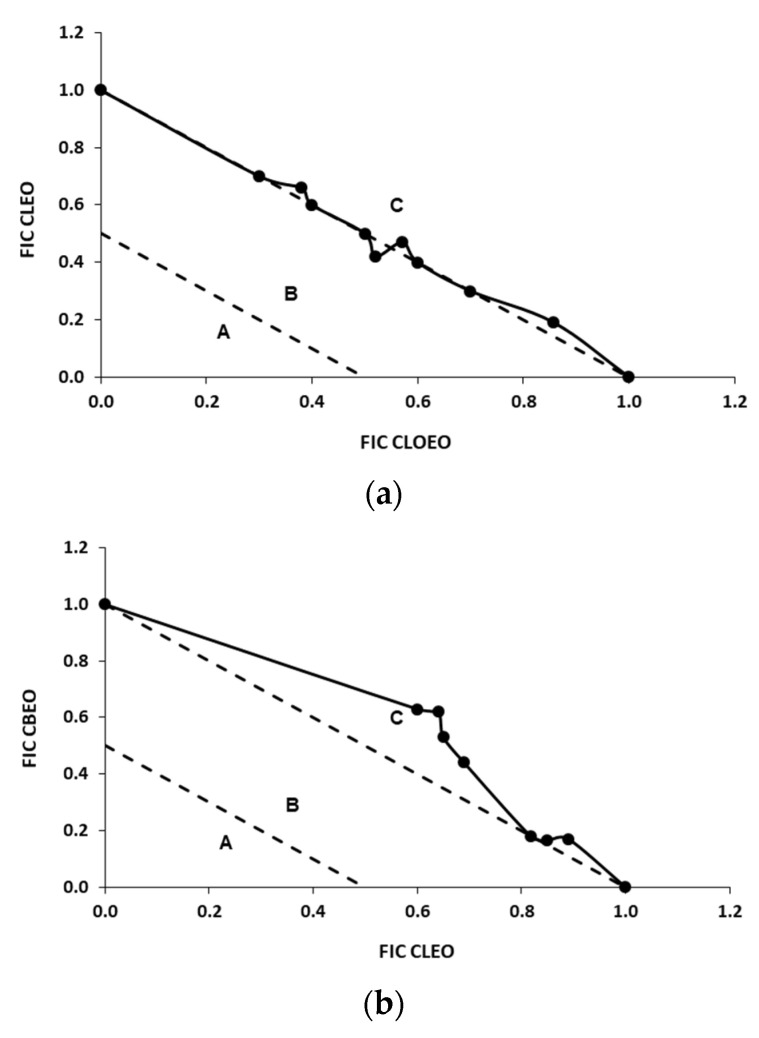
Isobolograms of combinations of EOs (CLEO, CBEO, and CLOEO) and pure compounds (VA and CA) against *E. coli* O157:H7. (**a**) CLEO/CLOEO; (**b**) CBEO/CLEO; (**c**) CBEO/CLOEO; (**d**) CLOEO/VA; (**e**) CLEO/VA; (**f**) CBEO/VA; (**g**) EU/CA. See Figure 1 for the interpretation of the isobolograms.

**Table 1 foods-10-01406-t001:** Antimicrobial activity of the tested antimicrobials against *L. monocytogenes* and *E. coli* O157:H7. MIC and MBC values are expressed in ppm.

	*L. monocytogenes*		*E. coli* O157:H7	
	MIC	MBC	MIC	MBC
CLEO	508	1000	843	1200
CLOEO	402	800	778	1200
CBEO	404	1200	721	1500
EU	562	1000	766	1200
CA	449	1400	455	1400
VA	3002	8000	2795	6000

**Table 2 foods-10-01406-t002:** FIC index (FICi) and interaction of selected binary combinations of antimicrobial compounds against *L. monocytogenes.* MIC values are expressed in ppm.

Combinations A/B	MIC_(A alone)_	MIC_(B alone)_	MIC_(A in the presence of B)_	MIC_(B in the presence of A)_	FICi
CLOEO/CLEO	402	508	121	280	0.85
CLEO/CBEO	508	404	153	162	0.70
CLOEO/CBEO	402	404	121	145	0.66
CLOEO/VA	402	3003	121	691	0.53
CLEO/VA	402	3003	189	691	0.67
CBEO/VA	404	3003	85	910	0.51
EU/CA	563	449	186	189	0.75

**Table 3 foods-10-01406-t003:** FIC index (FICi) and interaction of selected binary combinations of antimicrobial compounds against *E. coli* O157:H7. MIC values are expressed in ppm.

Combinations A/B	MIC_(A alone)_	MIC_(B alone)_	MIC_(A in the presence of B)_	MIC_(B in the presence of A)_	FICi
CLOEO/CLEO	779	844	335	414	0.92
CLEO/CBEO	844	721	658	166	1.00
CLOEO/CBEO	779	721	218	599	1.11
CLOEO/VA	799	2796	104	1006	0.49
CLEO/VA	844	2796	118	979	0.49
CBEO/VA	721	2796	144	784	0.48
EU/CA	767	455	421	45	0.65

## Data Availability

The data presented in this study are available on request from the corresponding author.

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
