# Peer review of "Synergistic Antimicrobial Activities of Combinations of Vanillin and Essential Oils of Cinnamon Bark, Cinnamon Leaves, and Cloves"

_foods, 2021, doi:10.3390/foods10061406_

Round 1

Reviewer 1 Report

Lines 173, 178 and throughout the work: Latin names for bacteria should be written in italics.

Lines 175 – 176 (Table 1): please specify the units of MIC and MBC.

Lines 296 – 297: correct the name of Salmonella Typhimurium, which is an abbreviation of the full name: Salmonella enterica subsp. enterica var. Typhimurium.

Lines 366 – 362: Is there any more scientific evidence for the phenomenon "combinations between different parts of a plant increase its antimicrobial effect" ?

Reviewer 2 Report

Investigation of antimicrobial effects of essential oils is an interesting area of food researches that can provide useful information not just for the science but also for the practice. Combination of vanillin with essential oils of cinnamon bark, cinnamon leaves and cloves can be a reliable option.

But, in my opinion, the novelty of the research presented in manuscript foods-1254039 is not given clearly. Unfortunately, Introduction section is too superficial. In my opinion, the results cannot be considered as complete to give enough details about the applicability of vanillin/cinnamon Eos.

Comments:

I suggest the authors to give the novelty of the study.

In my opinion the Abstract is too general ( in line 30: ‘some of the tested combinations increase the antimicrobial effect’, for instance).

In the Introduction section, the experiences related to the use of vanillin and OOs of cinnamon is not discussed in details (based on references).

In which food can apply the combination of vanillin/cinnamon? Please add this information to the Introductions and Discussion sections.

Figure 1 has very low quality.

Authors concluded in line 48 that EOs has’ higher costs compared with using synthetic agents’. But in the manuscript the costs are not discussed in details.

Authors mentioned in line 50 the ’ ..adverse effects after the EO treatment (e.g., changes in the physicochemical and sensory  characteristics..’. Indeed, the sensory properties have high relevance for food. But authors have not investigated the sensory properties.

Round 2

Reviewer 2 Report

The use of essential oils as antimicrobial agents for food is an interesting area of researches. Therefore, manuscript foods-1254039 can provide useful information for the science and practice, as well. Authors have revised the manuscript thoroughly according to reviewers’ comments and suggestions. In the revised form of manuscript the novelty of study has clearly given research motivations have defined well. Rephrasing of manuscript made it clear and more concrete. Introduction section has been amended by discussion based on additional references. References used for Introduction and results and discussion part are relevant. The qualities of figures have been improved after the revision. Rephrasing and additional information related to the economy of antimicrobial agents and sensory characteristics made the manuscript more complete. After the revision the overall scientific quality of manuscript has been improved significantly. I accept all answers and modifications made by the authors.